# `ZeroWaste` Dataset: Towards Automated Waste Recycling

## Abstract

Less than 35% of recyclable waste is being actually recycled in the US [1], which leads to increased soil and sea pollution and is one of the major concerns of environmental researchers as well as the common public. At the heart of the problem is the inefficiencies of the waste sorting process (separating paper, plastic, metal, glass, etc.) due to the extremely complex and cluttered nature of the waste stream. Automated waste detection strategies have a great potential to enable more efficient, reliable and safer waste sorting practices, but the literature lacks comprehensive datasets and methodology for the industrial waste sorting solutions. In this paper, we take a step towards computer-aided waste detection and present the first in-the-wild industrial-grade waste detection and segmentation dataset, `ZeroWaste`. This dataset contains over 1800 fully segmented video frames collected from a real waste sorting plant along with waste material labels for training and evaluation of the segmentation methods, as well as over 6000 unlabeled frames that can be further used for semi-supervised and self-supervised learning techniques. `ZeroWaste` also provides frames of the conveyor belt before and after the sorting process, comprising a novel setup that can be used for weakly-supervised segmentation. We present baselines for fully-, semi- and weakly-supervised segmentation methods. Our experimental results demonstrate that state-of-the-art segmentation methods struggle to correctly detect and classify target objects which suggests the challenging nature of our proposed in-the-wild dataset. We believe that `ZeroWaste` will catalyze research in object detection and semantic segmentation in extreme clutter as well as applications in the recycling domain. Our project page can be found at http://ai.bu.edu/zerowaste/.

## 1   Introduction

As the world population grows and gets increasingly urbanized, waste production is estimated to reach 2.6 billion tonnes a year in 2030, an increase from its current level of around 2.1 billion tonnes [5]. Efficient recycling strategies are critical to reduce the devastating environmental effects of rising waste production. Materials Recovery Facilities (MRFs) are at the center of the recycling process. These facilities are where the collected recyclable waste is sorted into separate bales of plastic, paper, metal and glass. The accuracy of the sorting directly determines the quality of the recycled material; for high-quality, commercially viable recycling, the contamination levels (anything but the desired material) need to be less than a few percent of the bale. Even though the MRFs utilize a large number of machinery alongside manual labor [6], the extremely cluttered nature of the waste stream makes automated waste detection (*i.e.* detection of waste objects that should be removed from the conveyor belt) very challenging to achieve, and the recycling rates as well as the profit margins stay at undesirably low levels (e.g. less than 35% of the recyclable waste actually got recycled in the United States in 2018 [1]). Another crucial aspect of manual waste sorting is the safety of the workers that risk their lives daily picking up unsanitary objects (*e.g.* medical needles).

Submitted to 35th Conference on Neural Information Processing Systems (NeurIPS 2021). Do not distribute.

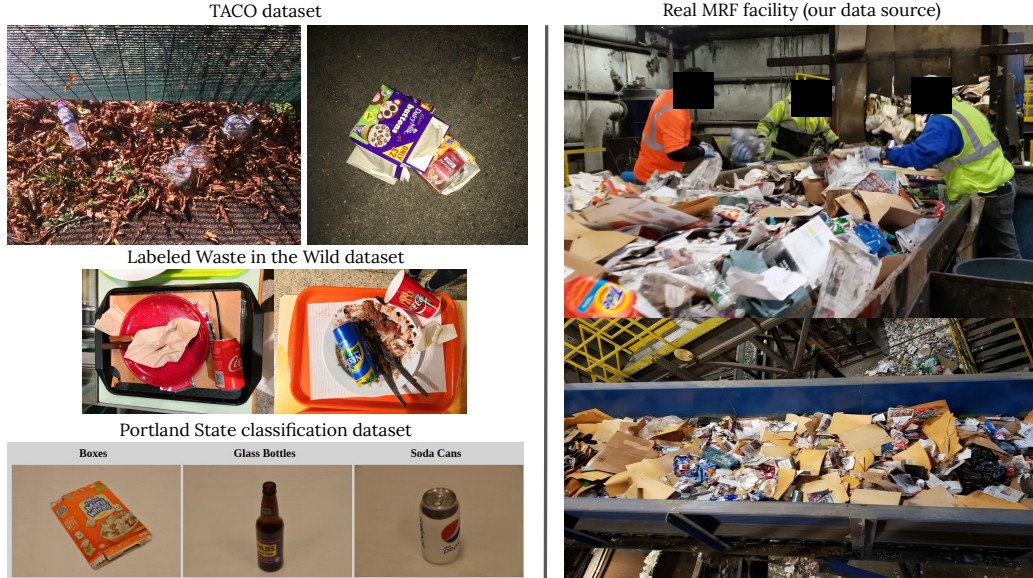

Figure 1: **Left:** examples of the existing waste detection and classification datasets (top to bottom): Trash Annotation in Context (TACO) [2], Labeled Waste in the Wild [3], Portland State University Recycling [4] datasets. **Right:** footage of the waste sorting process at a real Materials Recovery Facilities (MRF). The domain shift between the simplified datasets with solid background and little to no clutter and the real images of the conveyor belt from the MRF makes it impossible to use models trained on these datasets for automated detection on real waste processing plants. In this paper, we propose a new `ZeroWaste` dataset collected from a real waste sorting plant. Our dataset includes a set of densely annotated frames for training and evaluation of the detection and segmentation models, as well as a large number of unlabeled frames for semi- and self-supervised learning methods. We also include frames of the conveyor belt before and after manual collection of foreground objects to facilitate research on weakly supervised detection and segmentation. Please see Figure 2 for the illustration of our `ZeroWaste` dataset.

Recent advances in object classification and segmentation provide a great potential to make the recycling process more efficient, more profitable and safer for the workers. Accurate waste classification and detection algorithms have a potential to enable new sorting machinery (e.g. waste sorting robots), improve the performance of existing machinery (e.g. optical sorters [6]), and allow automatic quality control of the MRFs' output. Unfortunately, the research community is lacking the gold-standard in-the-wild datasets to train and evaluate the classification and segmentation algorithms for industrial waste sorting. While several companies do development on this subject (e.g. [7, 8, 9]), they keep their dataset private, and the few existing open-source datasets [10, 4, 3, 2] are very limited in data amount and/or generated in uncluttered environments, not representing the complexity of the domain (see Figure 1). In this paper, we propose a first large-scale in-the-wild waste detection dataset `ZeroWaste` that is specifically designed for the industrial waste detection. `ZeroWaste` is a dataset that is fundamentally different from the popular detection and segmentation benchmarks: high level of clutter, visual diversity of the foreground and background objects that are often severely deformed, as well as a fine-grained difference between the object classes (*e.g.* brown paper vs. cardboard, soft vs. rigid plastic) – all these aspects pose a unique challenge for the automated vision. We envision that our open-access dataset will enable computer vision and robotics communities to develop more robust and data-efficient algorithms for object detection, robotic grasping and other related problems.

Our contributions can be summarized as follows:

1. We propose the first fully-annotated `ZeroWaste-f` dataset specifically designed for industrial waste object detection. The proposed `ZeroWaste-f` dataset contains video frames from a real MRF conveyor belt densely annotated with instance segmentation and proposes a challenging real-life computer vision problem of detecting highly deformable objects in severely cluttered scenes. In addition to the fully annotated frames from `ZeroWaste-f` set, we include the unlabeled `ZeroWaste-s` set for semi-supervised learning.

2. We introduce a novel before-after data collection setup and propose the `ZeroWaste-w` dataset for binary classification of frames before and after the collection of target objects. This binary

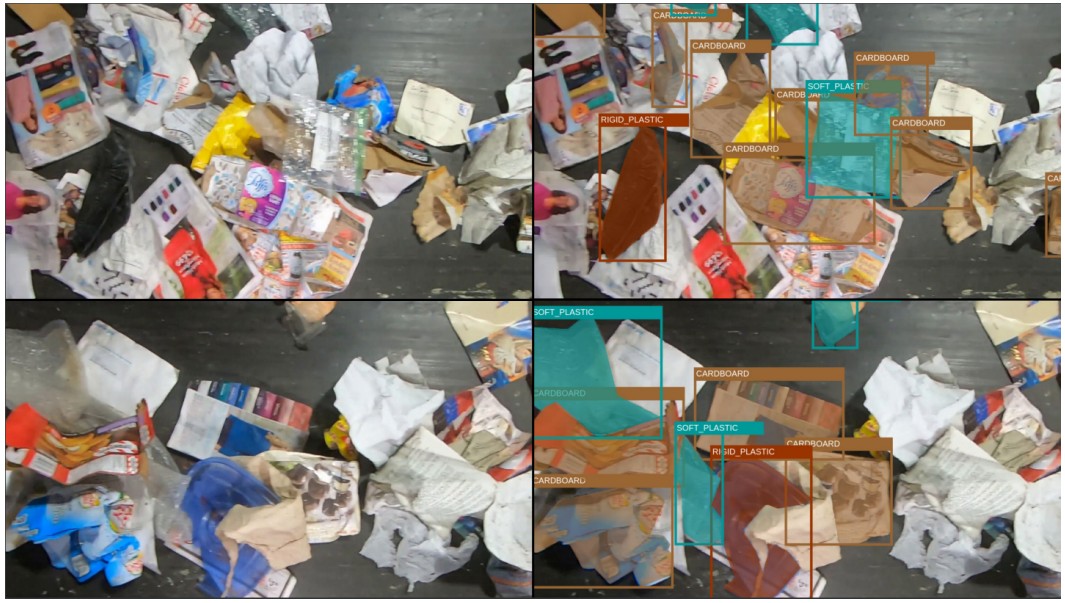

Figure 2: Examples of images (**left**) and the corresponding polygon annotation (**right**) of the proposed `ZeroWaste` dataset. At the end of this conveyor belt, only paper objects must remain. Therefore, we annotated the objects of four material types that should be removed from the conveyor belt as foreground: soft plastic, rigid plastic, cardboard and metal. The background includes the conveyor belt and paper objects. Severe clutter and occlusions, high variability of the foreground object shapes and textures, as well as severe deformations of objects usually not present in other segmentation datasets, make this domain very challenging for object detection. More examples of our annotated data can be found in Section B.3 of the Appendix (*best viewed in color*).

classification setup allows much cheaper data annotation and allows further development of weakly supervised segmentation and detection methods.

3. We implement the fully-supervised detection and segmentation baselines for the `ZeroWaste-f` dataset and semi- and weakly-supervised baselines for `ZeroWaste-s` and `ZeroWaste-w` datasets. Our experimental results show that popular detection and segmentation methods struggle to generalize to our proposed data, which indicates a challenging nature of our in-the-wild dataset and suggests that more robust and data-efficient methods must be developed to solve the waste detection problem.

## 2 Related Work

**Detection and Segmentation Datasets**    Many datasets for image segmentation have been proposed with the goal of densely recognizing general objects and "stuff" in image scenes like street view [11, 12, 13], natural scenes [14, 15, 16, 17, 18], and indoor spaces [19, 20, 21]. Yet, few of them have been designed for the more challenging vision task required in automated waste recycling, aiming to densely identify and segment deformable recyclable materials, many of which look very similar to each other, from a highly cluttered background [6]. Several related datasets have been proposed that contain only image-level labels. For example, *Portland State University Recycling* [4] consists of 11500 labeled images of five common recyclable types: box-board, glass bottles, soda cans, crushed soda cans and plastic bottles. Similarly, *Stanford TrashNet* [10] presents 400 images containing a single waste object from six predefined classes. Though beneficial for image-level classification in well-defined conditions, images of in these two datasets have very simple background and do not apply to waste object localization. To enable localization tasks, *Labeled Waste in the Wild* [3] annotated bounding boxes for objects of 20 classes in 1002 food tray photos. *Annotation in Context (TACO)* [2] went one step further by densely annotating 60 litter objects from 1500 images. Yet TACO contains deliberately collected outdoor scenes with one or a few foreground objects that are rarely occluded, which makes it less practical for materials recovery scenarios. In contrast, our `ZeroWaste` was collected from the front lines of a waste sorting plant where the collected objects

are frequently severely deformed and occluded, which makes both detection and segmentation a significantly more challenging and practical task.

**Detection and Segmentation Methods**  Image segmentation is an essential component in robotic systems like automated waste sorters [6], as it partitions images into multiple regions or objects suitable for grasping. Image segmentation can be formulated as a task that classifies each pixel into a set of labels [22]. Recent semantic segmentation models [23, 24, 25, 26] have achieved state-of-the-art performance for recognizing general object/stuff classes from natural scene images. Instance segmentation [27, 28, 29, 30] works by further consider instance identity for objects. Representative frameworks like MaskRCNN [31] effectively detect objects in images and simultaneously generate high-quality masks, which enables efficient interaction between robots and target objects. Yet due to their data-hungry nature, these methods rely on large volumes of annotated data for training, which can be challenging and expensive, especially in specialized application scenarios [32]. Recycling annotation in particular requires expert labelers and is thus even more costly. Semi-supervised segmentation methods have been proposed to address such limitations by jointly learning from both annotated and unannotated images [33, 34, 35, 36, 37, 38]. Weakly-supervised segmentation methods exploit annotations that are even easier to obtain, e.g. image-level tags [39, 40, 41]. These methods typically utilize class activation maps (CAM) [42] to select the most discriminative regions, which are later used as pixel-level supervision for segmentation networks [43, 44, 45]. All these advanced segmentation models are trained on general-purpose data, and applying them to waste sorting scenarios presents challenges like domain shift. To study the effectiveness of existing models and enable further improvement for the waste sorting task, we test our proposed `ZeroWaste` with previous state-of-the-art methods and report their performance as baselines.

# 3  `ZeroWaste` Dataset

In this section, we describe our `ZeroWaste-f` dataset for fully supervised detection and evaluation, unlabeled `ZeroWaste-s` data for semi-supervised learning and `ZeroWaste-w` dataset of images before and after the removal of target objects for weakly supervised detection. The datasets are licensed under the Creative Commons Attribution-NonCommercial 4.0 International License [46]. The MRF at which the data was collected agreed to release the data for any non-commercial purposes and decided to remain unacknowledged.

**Data Collecion and Pre-processing**  The data was collected from a high-quality paper conveyor of a single stream recycling facility in Massachusetts. The sorting operation on this conveyor aims to keep high quality paper and consider anything else as contaminants including non-paper items (*e.g.* metal, plastic, brown paper, cardboard, boxboard). We collected data during the regular operation of the MRF using two compact recording installations at the start and end of the conveyor belt (see Fig. 3, right), that is, footage is captured simultaneously both at the unsorted and sorted sections of the same conveyor. The recording apparatus is designed to fit the constraints of the facility: In order not to disrupt the MRF operation and be able to work in confined spaces available near the conveyor the recording platform needs to be compact, non-intrusive (to the workers), and portable (easy to move, battery-powered). Note that the cameras are not directly mounted on the conveyor but to a stand-alone platform, to reduce vibrations transmitted to the cameras. Additional considerations are made (see Figure 3, center): (1) Damping pads are installed to counter the ground vibrations of the heavy machinery and reduce vibrations on the camera even further; (2) Weighted bases lower the center of mass to keep the apparatus stable.

We used the GoPro Hero 7 for RGB footage, and we additionally collected the the near-infrared (NIR) footage simultaneously with the RGB footage using the MAPIR Survey3W NIR camera for the future work (specifically, it captures at a wavelength of 850 nm). The cameras in their encasings meet both the portability and ruggedness requirements. To maintain consistent lighting, two LitraTorch 2.0 portable lamps are installed with a light diffuser. This softens the light and spreads it more evenly in the scene. Both cameras were installed at around 100 cm above the conveyor, and the light sources at around 80 cm. Sequences of twelve videos of total length of 95 minutes and 14 seconds with FPS 120 and size $1920 \times 1080$ were collected and processed. The preprocessing of the collected data involved the following steps:

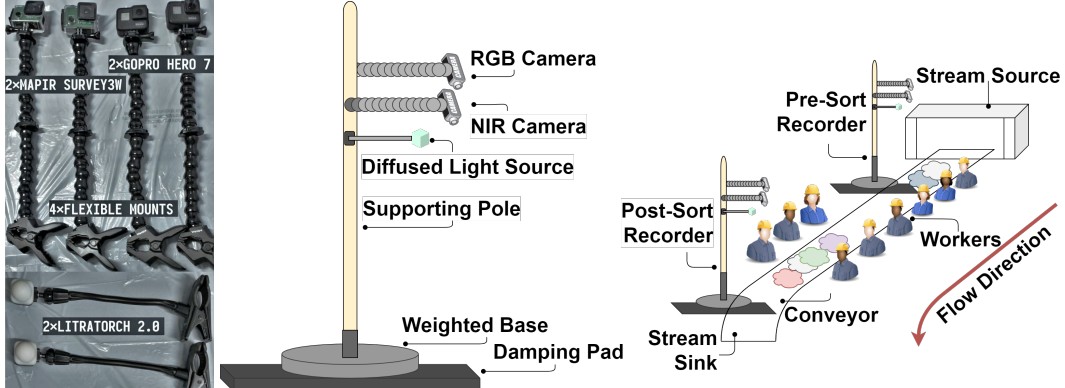

Figure 3: The footage recording setup is designed to fit the constraints of the facility environment. **Left:** The specific cameras and lamps used. **Center:** Assembly of each recording apparatus. **Right:** Layout of the recording setup in the recycling environment.

image          ground truth

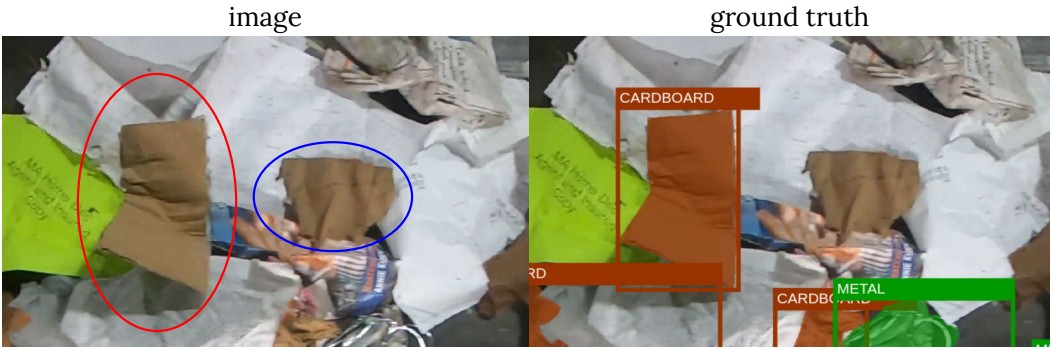

Figure 4: **Left:** example of an image from `ZeroWaste-f` dataset. **Right:** the corresponding ground truth instance segmentation. Expert training and common sense knowledge are required to distinguish between the cardboard object on the left (red circle) and the brown paper on the right (blue circle), as they are visually very similar but differ in thickness and rigidity (*best viewed in color*).

1. Rotation and cropping. The frames were rotated so that the conveyor belt is parallel to the frame borders and cropped to remove the regions outside the conveyor belt. We ensured that any personal information or identifiable footage of the workers at the conveyor belt was excluded from our data.
2. Optical distortion. We removed the distortion [47] using the OpenCV [48] library to compensate for the fish-eye effect caused by the proximity of the cameras to the conveyor belt.
3. Deblurring. We used the SRN-Deblur [49] method to remove motion blur resulting from the fast-moving conveyor belt. According to our visual inspection, SRN-Deblur achieves satisfactory deblurring and does not introduce the undesired artifacts that usually appear when classical deconvolution-based methods are used.
4. Subsampling. We sampled every tenth frame from the video to avoid redundancy.

The illustration of the original frames shot at the beginning of the conveyor belt and the corresponding preprocessing results can be found on Figure 8 in Section B.3 of the Appendix.

**Densely Annotated `ZeroWaste-f` and Unlabeled `ZeroWaste-s` Datasets** The fully annotated `ZeroWaste-f` dataset consists of 1874 frames sampled from the processed videos and the corresponding ground truth polygon segmentation. We used the open-source CVAT [50] annotation toolkit to manually collect the polygon annotations of objects of four material types: cardboard, soft plastic, rigid plastic and metal. We chose this set of class labels following the MRF's guidelines for the workers to collect cardboard, plastic and metal into separate bins, as well as the fact that grasping of rigid and non-rigid objects might require the use of fundamentally different kinds of robotic systems. The polygon annotation was performed according to the following set of rules:

| Split | #Images | Carboard | Soft Plastic | Rigid Plastic | Metal | #Objects |
|---|---|---|---|---|---|---|
| Train | 1245 | 4038 | 1550 | 460 | 114 | 6162 |
| Validation | 312 | 795 | 310 | 195 | 24 | 1324 |
| Test | 317 | 1216 | 466 | 242 | 53 | 1977 |
| Unlabeled | 6212 | - | - | - | - | - |
| Total | 8086 | 6049 | 2326 | 897 | 191 | 9463 |

Table 1: Statistics of the training, validation and test splits of our `ZeroWaste-f` dataset *w.r.t.* the number of labeled objects, and the additional unlabeled `ZeroWaste-s` set of images for semi-supervised learning.

1. Objects of four material types were annotated as foreground: cardboard (including parcel packages, boxboard such as cereal boxes and other carton food packaging), soft plastic (*e.g.* plastic bags, wraps), rigid plastic (*e.g.* food containers, plastic bottles) and metal (*e.g.* metal cans). Paper objects were treated as background.
2. The entire object must be within the corresponding polygon.
3. If an object is partially occluded and separate parts are visible, we annotated them as separate objects.

Each annotated video frame was validated by an independent reviewer to pass the standards above (see Figure 2). Both the annotation and the review process were performed by the students and researchers with a computer science background specifically trained to perform the annotation. We did not delegate the annotation to the crowd-sourcing platforms, such as Amazon Mechanical Turk [51], due to the complexity of the domain that requires expert knowledge to be able to detect and correctly classify the foreground objects (see the illustration on Figure 4). The estimated average cost of the annotation and review is about 12.5 minutes per frame. The dataset was split into training, validation and test splits and stored in the widely used MS COCO [18] format for object detection and segmentation using the open-source Voxel51 toolkit [52]. Please refer to Table 1 for more details about the class-wise statistics of all splits. In addition to the fully annotated `ZeroWaste-f` examples, we provide 6212 unlabeled images that can be used to refine the detection using semi-supervised or self-supervised learning methods. We refer to this unlabeled set of images as `ZeroWaste-s` data later on in this paper.

**`ZeroWaste-w` Dataset for Binary Classification** We leverage the videos taken of the conveyor belt before and after the removal of the foreground objects to create a weakly-supervised `ZeroWaste-w` dataset. This dataset contains 1202 frames with the foreground objects (*before* class) and 1208 frames without the foreground objects (*after* class). One advantage of such a setup is that it is relatively cheap to acquire the ground truth labels (only an image-level inspection is required to ensure there are no false negatives in the *after* class subset). The `ZeroWaste-w` dataset is specifically collected to be used in the weakly-supervised setup and is meant to provide an alternative and more data-efficient solution to the problem. The ground truth instance segmentation is available for all images of the *before* class as it overlaps with the `ZeroWaste-f` dataset. Please see Figure 5 for an illustration of the `ZeroWaste-w` examples.

## 4 Experiments

In this section, we provide baseline results for our proposed `ZeroWaste` dataset. We perform fully supervised instance and semantic segmentation on `ZeroWaste-f` using the most widely used Mask R-CNN [31] and DeepLabV3+ [53] respectively. We also perform fully- and semi-supervised semantic segmentation on `ZeroWaste-s` using the CCT [33] method, and report the initial segmentation quality of CAMs produced by a classifier trained on `ZeroWaste-w` dataset as a weakly-supervised baseline. The implementation of our experiments and the detailed description of the experimental setup are available at https://github.com/dbash/zerowaste.

### 4.1 Object Detection

**Experiments with COCO-pretrained Networks** It has been shown that pretraining the model on a large-scale dataset, such as MS COCO [18], improves generalization and helps to prevent severe overfitting in case when the target dataset is relatively small [54, 55, 56]. Therefore, in our

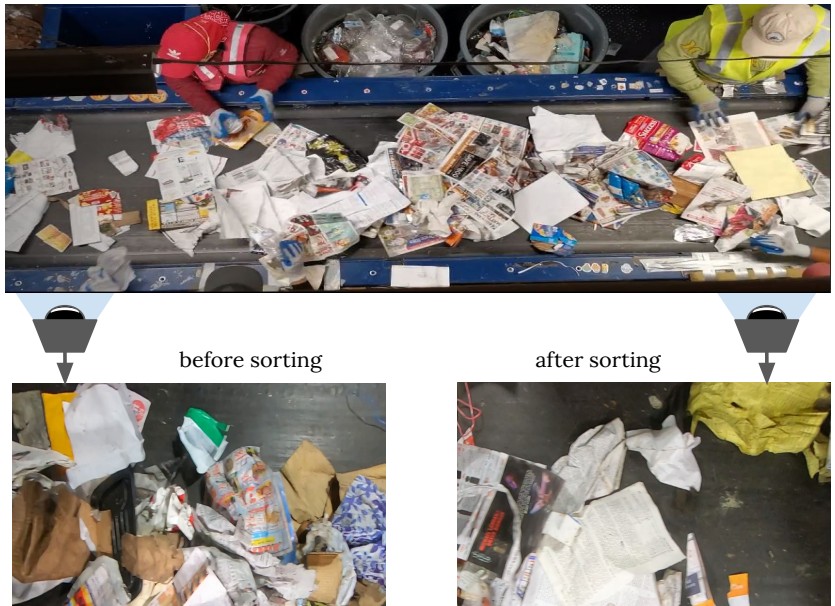

before sorting          after sorting

Figure 5: We installed two stationary cameras above the conveyor belt: one at the beginning of the line and another one at the end. At this particular conveyor belt, workers are asked to remove objects of any material other than paper, such as cardboard, plastic and metal. Therefore, the footage collected from the beginning of the line contains the "foreground" objects that need to be removed, and the frames from the end of the conveyor belt are supposed to only contain the "background" paper objects. We used this setup as a foundation of our `ZeroWaste-w` dataset.

first experiments, we used the initialized the model with weights learned on COCO and further finetuned it with our `ZeroWaste-f` dataset. We used a standard implementation of the popular Mask R-CNN with ResNet-50 [57] backbone provided in the popular Detectron2 [58] library in all of the experiments. The model was finetuned for 40000 iterations on the training set of our `ZeroWaste-f` dataset on a single Geforce GTX 1080 GPU with batch size 8. To compensate for a relatively small number examples in the training set and to avoid overfitting, we leveraged heavy data augmentation, including random rotation and cropping, adjustment of brightness and hue, *etc.* We report the experimental results in Table 2 (COCO → `ZeroWaste` section). A more detailed description of the results can be found in Section B.1 of Appendix.

**Experiments with TACO-pretrained Mask RCNN** In the next set of experiments, we utilize the TACO dataset for waste detection in the outdoor scenes distributed under Attribution 4.0 International (CC BY 4.0) license. We trained Mask R-CNN for 40000 epochs on the modified TACO dataset with the material-based labels (cardboard, soft plastic, rigid plastic, metal and other) initialized with weights from MS COCO. We then finetuned the model on the training set of `ZeroWaste-f` data and report the results in Table B.1 (TACO→ `ZeroWaste` section).

**Results** The experimental results with Mask RCNN indicate severe overfitting to the training data, hence the model fails to generalize to the unseen examples. The model pretrained on the TACO dataset performs poorly on both TACO and `ZeroWaste-f` datasets, which shows that, despite its remarkable efficiency on the large-scale datasets with natural scenes, such as MS COCO or Pascal VOC [59], Mask RCNN cannot generalize to our relatively small, extremely cluttered data with very diverse deformable objects. Recalling the history of success with other complex segmentation and detection datasets (*e.g.* from mIoU 57% in 2015 [60] to 84% in 2020 [61] on CityScapes [11], or from 51.6% in 2014 [62] to 90% in 2020 [63] on PASCAL VOC 2012 [59]), and knowing that the task *can* be solved by humans with a little training, we believe that the computer vision community will eventually come up with efficient methods for this challenging task.

### 4.2 Semantic Segmentation

**Fully supervised experiments** We used the state-of-the-art DeeplabV3+ model as a fully-supervised semantic segmentation baseline for our dataset. DeeplabV3+ is an efficient segmentation model that combines the atrous convolutions to extract the features in multiple scales, and an encoder-

|  | TACO → `ZeroWaste` | | | COCO → `ZeroWaste` | | |
|---|---|---|---|---|---|---|
|  | AP | AP50 | AP75 | AP | AP50 | AP75 |
| **Train** | 39.11 | 54.77 | 44.58 | 62.55 | 81.59 | 71.59 |
| **Validation** | 15.86 | 28.83 | 16.37 | 14.99 | 23.62 | 16.09 |
| **Test** | 14.55 | 25.9 | 14.81 | 14.79 | 25.94 | 14.82 |

Table 2: Instance segmentation results of Mask R-CNN pretrained on TACO dataset (**left**) and MS COCO dataset (**right**).The model pretrained on MS COCO overfits to the training split, while pretraining on TACO dataset significantly reduces overfitting but does not yield a significant improvement in detection accuracy on the validation and test sets. Please refer to Tables 6 in the Appendix for class-wise results.

decoder paradigm to gradually sharpen the object boundary using the intermediate features. As in the detection experiments, we used a standard implementation of DeeplabV3+ from Detectron2 library. We used the model with ResNet-101 backbone with three $3 \times 3$ convolutions instead of the first $7 \times 7$ convolution that was pretrained on Cityscapes dataset [11]. We froze the first three stages of the backbone (convolution and two first residual block groups) and finetuned the model on the training set of `ZeroWaste-f` for 10000 iterations with starting learning rate 0.01 and batch size 40 on a single GPU RTX A6000 which took approximately 14 hours. As in the previous experiments, we augmented the data extensively to prevent overfitting. The results of our experiments on all `ZeroWaste-f` splits can be found on the Table 3.

**Semi-supervised experiments** For a semi-supervised segmentation baseline, we used an official implementation of Cross-Consistency Training CCT [33] method. CCT uses a shared encoder and several auxiliary decoders each of which performs various augmentations, such as spatial dropout, random noise, cutout of object regions *etc.* , and a cross-entropy-based loss to force the unlabeled predictions to be consistent across all decoders. Since CCT uses a different backbone architecture from DeeplabV3+, we first trained CCT on the labeled `ZeroWaste-f` data only for comparison with the semi-supervised setting. We used the same default hyperparameters reported in the paper for both supervised and semi-supervised experiments (the exact configuration can be found in our project). We report the mean Intersection over Union (mIoU) as well as mean pixel accuracy for both setups in Table 3, and more details can be found in Section B.2 of the Appendix.

**Weakly-supervised baseline** As a baseline for weakly-supervised segmentation, we trained a binary classifier on the before and after collection frames of the `ZeroWaste-w` dataset. We used a standard Pytorch [64] implementation of ResNet50 [57] pretrained on ImageNet [65] for our classifier, and trained it for 5 epochs with learning rate $5 \times 10^{-4}$ using the binary cross-entropy loss. The resulting classifier obtained over $98\%$ accuracy on the test set. We then used RISE [66], a black-box saliency generating technique, to extract the class activation maps (CAMs). RISE masks the input image with a set of random binary masks and returns the linear combination of the resulting CAMs weighted with the corresponding masks. The maps generated by RISE are then normalized and thresholded with 0.621 that results in highest mIoU on the training set. For comparison, we computed the mean pixel accuracy and mIoU on randomly generated masks with the probability of each pixel belonging to the foreground class equal to the average fraction of the foreground pixels in the `ZeroWaste-w` dataset $14.9\%$ and report these results in Table 3. The visualization of the resulting CAMs can be found in Figure 9 in Section B.3 of the Appendix.

**Results** Experimental results in Table 3 indicate that our `ZeroWaste` dataset proposes a challenging semantic segmentation task with an unusual for the standard segmentation datasets level of clutter, diversity of the foreground objects and, at the same time, their visual similarity with the background objects (all methods often tend to mistake the paper objects for cardboard and vice versa, and have a hard time distinguishing between soft and rigid plastic objects). The semi-supervised learning results indicate that the unlabeled examples from the `ZeroWaste-s` subset do not significantly help CCT improve the overall segmentation quality. As seen from the class-wise segmentation results on Table 8 in Section B.2 of Appendix, additional training of CCT with unlabeled data results in higher segmentation accuracy of the most frequent classes (*e.g.* cardboard and background), but degrades the performance on the objects of the rare classes (*e.g.* metal). Additionally, the binary classification results show that a simple CAM-based approach with cheap `ZeroWaste-w` data provides meaningful localization cues that can be further used for weakly- and semi-supervised segmentation.

| | Supervision | Train | | Validation | | Test | |
|---|---|---|---|---|---|---|---|
| | | mIoU | Pixel Acc. | mIoU | Pixel Acc. | mIoU | Pixel Acc. |
| *Random* | none | 7.2 | 74.7 | 7.2 | 75.3 | 8.4 | 71.8 |
| *CAM* | weak | 15.7 | 43.9 | 16.3 | 47.5 | 18.6 | 43.2 |
| *CCT semi* | semi | 61.2 | 97.4 | 29.40 | 83.3 | 30.0 | 83.6 |
| *CCT* | full | 65.38 | 97.9 | 29.80 | 83.4 | 29.20 | 81.2 |
| *DeeplabV3+* | full | 88.5 | 98.19 | 40.16 | 91.23 | 39.06 | 88.47 |

Table 3: Results of CAMs produced by RISE [66] with a binary classifier trained on `ZeroWaste-w` before and after frames, CCT [33] trained only using the `ZeroWaste-f` , CCT trained with `ZeroWaste-f` and `ZeroWaste-s` , and DeepLabV3+ [53] on our `ZeroWaste-f` dataset. Results indicate that 1) severe overfitting occurs in the supervised scenario; 2) unlabeled `ZeroWaste-s` images do not significantly improve the segmentation quality of CCT and 3) the binary classifier trained on `ZeroWaste-w` provides plausible localization guidance that that can serve as cues for weakly-supervised segmentation. Please refer to Tables 7 and 8 for class-wise segmentation results and Figure 7 in the Appendix for confusion matrices on all splits.

## 5 Impact and Limitations of `ZeroWaste`

**Machine Learning Research** `ZeroWaste` provides a gold standard for the evaluation of different waste sorting methods. It will catalyze research in the areas of fully, semi, and weakly supervised segmentation, data-efficient learning and domain adaptation. Our dataset provides a real-world application that is significantly more challenging than the previously used benchmarks for these tasks.

**Robotics Research** This dataset will enable the development of robotic manipulation algorithms for waste sorting. It will facilitate research in object picking algorithms that can work with extremely cluttered scenes using realistic segmentation polygons. Integrating high-level reasoning about object classes and properties (e.g. hard/soft materials) to the picking algorithm will provide novel research avenues and can significantly boost the picking accuracy.

**Limitations and Future Directions** Despite the fact that `ZeroWaste` is to the date the largest public dataset for waste detection and segmentation, it is still smaller than the standard large-scale benchmarks due to the fact that the annotation process for this domain is very expensive. For this reason, state-of-the-art detection and segmentation methods tend to overfit to the training data and therefore do not generalize well to the unseen examples. As future work, we plan to increase the diversity of our dataset by using synthetic-to-real domain adaptation and other data augmentation techniques. Another important future direction is to utilize visual signals of other modalities, *e.g.* near infrared footage that can be especially useful for distinguishing different material types.

**Societal Impact** This paper is a part of a collaboration project that investigates the implications of deploying new AI and Robotics algorithms to MRFs [67]. We believe that human-robot collaboration is essential for more efficient computer-aided recycling, quality control of the sorting process, as well as in establishing safer work conditions for the MRF workers (*e.g.* by detecting dangerous waste items, such as sharp or explosive objects). This dataset can potentially be used to develop fully-automated MRFs with waste sorting robots, which may compromise the financial security of the MRF workers. However, after consulting with experts, we found that such fully-automated solutions would be far from sufficient to meet the contamination levels required in recycling, especially considering the complex, cluttered and varying nature of the waste stream. Given that only a small portion of the recyclable waste is currently getting recycled, achieving an efficient human-robot collaboration has a potential to solve the pressing problem of water and soil pollution.

## 6 Conclusion

This work introduces the largest public dataset for waste detection. `ZeroWaste` is designed as a benchmark for training and evaluation of fully, weakly, and semi-supervised detection and segmentation methods, and can be directly used for a broader category of tasks including transfer learning, domain adaptation and label-efficient learning. We provide baseline results for the most popular fully, weakly, semi-supervised, and transfer learning techniques. Our results show that current state-of-the-art detection and segmentation methods cannot efficiently handle this complex in-the-wild domain. We anticipate that our dataset will motivate the computer vision community to develop more data-efficient methods applicable to a wider range of real-world problems.

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
