# OpenReview forum: "ZeroWaste Dataset: Towards Automated Waste Recycling"
_NeurIPS.cc/2021/Track/Datasets_and_Benchmarks/Round1 — Submitted to NeurIPS 2021 Datasets and Benchmarks Track (Round 1)_

### Official Review · Reviewer_3fEG · 2021-07-04
**A good initiative for waste recycling**

**Rating:** 6
**Confidence:** 3
**Correctness:** yes
**Clarity:** yes

**Strengths:**

While the dataset has lots of environmental meaning, the authors also provide comprehensive evaluation on different baseline algorithms





**Weaknesses:**

The dataset is a good initiative, however, the task of object detection from MRF is very hard. From the images, we can see there are lots of occlusions, partial objects, noisy background, uneven lighting conditions etc. And the labels only cover the complete objects, which reduced the usability of the dataset. And the big gaps between training and test are very concerning.

**Additional Feedback:**

no

**Documentation:**

yes

**Ethics:**

There might be private information in the dataset (e.g. bank statement)

**Relation To Prior Work:**

yes

**Summary And Contributions:**

The paper proposed 3 datasets including ZeroWaste-f dataset that is fully-annotated for industrial waste object detection; unlabeled ZeroWaste-s set for semi-supervised learning, and ZeroWaste-w dataset for binary classification. Several baseline object detection and semantic segmentation algorithms are evaluated on the proposed datasets.

---

> ### Author Response · Authors · 2021-07-09
> **Reply to Reviewer 3fEG**
>
> We thank the Reviewer for their valuable feedback.
>
> *Task is too hard*: It is indeed a very challenging task, but very real at the same time. The fact that state-of-the-art methods do not perform well on this task shows that the CV community needs to develop methods that are more data-efficient and robust to clutter and class imbalance in order to be applicable to such real-world applications.
>
> *“And the labels only cover the complete objects”*:
> It is not clear to us what is precisely meant by this. The annotation includes all foreground objects, including partially occluded and partially out-of-view objects (lines 169-170). Line 168 alludes to the fact that the polygon might contain a little bit of background, but there should be no foreground object pixels outside the polygon.
>
> *“And the big gaps between training and test are very concerning”* :
> There is a performance gap between the train and test splits, which indicates overfitting. We believe that the state-of-the-art methods are targeted towards the domains where much more training data is available, which is prohibitively expensive in our domain. We argue that the only viable option is to develop better semi- and weakly-supervised methods, or a combination of such supervision signals, for which much more data is available.

---

### Official Review · Reviewer_akrh · 2021-07-04
**Waste detection and segmentation dataset for automated waste recycling**

**Rating:** 5
**Confidence:** 4
**Correctness:** Yes. The dataset is constructed in a …
**Clarity:** Yes. The paper is well written and ea…

**Strengths:**

The proposed dataset is meaningful for more efficient waste recycling, quality control of the sorting process and establishing safer work conditions for the MRF workers.

**Weaknesses:**

The novelty of this paper is limited, since it only conducts experiments with current existing state-of-the-art approaches.

**Additional Feedback:**

-

**Documentation:**

Yes.

**Ethics:**

Yes. This work can help for more efficient computer-aided recycling, quality control of the sorting process, as well as establishing safer work conditions for the MRF workers.

**Relation To Prior Work:**

Yes.

**Summary And Contributions:**

This paper presents a in-the-wild industrial-grade waste detection and segmentation dataset called ZeroWaste, which can serve for fully, weakly and semi-supervised detection and segmentation tasks.

The authors present baselines for fully, semi and weakly-supervised segmentation methods.

---

> ### Author Response · Authors · 2021-07-09
> **Reply to Reviewer akrh: limited novelty**
>
> *Novelty*: We would like to stress that this is a datasets track, so the main novelty of the submission lies in the new dataset we propose. To summarize, our contributions are: 1) We collected the first in-the-wild dataset for industrial waste sorting that is a crucial part of our ultimate goal of creating an open-source robot-assisted waste sorting system; 2) We annotated instance and semantic segmentation of over 1800 frames of the conveyor belt, which is sufficient to evaluate the supervised detection and segmentation methods and to train the semi-supervised methods; 3) In addition to the fully annotated and unannotated data, we provide a set of clean frames before and after the manual collection of foreground objects, which comprises a labeled dataset for weakly-supervised segmentation methods; 4) We evaluate the state-of-the-art detection and segmentation methods on our data in four setups: fully-supervised detection, fully-supervised semantic segmentation, semi-supervised and weakly-supervised semantic segmentation.

---

### Official Review · Reviewer_Criz · 2021-07-04
**An annotated dataset for waste sorting with baselines for segmentation and detection task is constructed.**

**Rating:** 5
**Confidence:** 4

**Strengths:**

-- Real data (in the wild) for waste sorting task
-- Dataset will be of interest to machine vision and robotics community
-- Accessibility is open-source for non-commercial use
-- Addresses an important application area


**Weaknesses:**

While I appreciate the efforts in the dataset construction for this important problem, there are a few points that are not very clear:

The image data is complex enough with variability in clutter, material types, occlusion. However, given that the problem involves material classification, isn't material classification usually done better by using hyperspectral imaging, use of polarization and other special sensors?  The authors mention the acquisition using NIR camera but as far as I can tell the benchmark data is only done on RGB data.   The authors claim that the datasets can be used for robotic grasping applications but I can't follow how the dataset in its present form is useful to robotics practitioners.  While I see potential extensions to the data collection process that may facilitate robotics research the current statement is an overclaim.

The sampling strategy used for the images is not clear - no mention of the conveyor belt speed is made to get an understanding of the degree of redundancy between subsequent frames in the video.

The authors claim that this is large scale dataset. While the number of images is in the order of 1000's, the complexity of the application may necessitate sampling across several hours of operation, with diverse variations in waste.  It is not clear how the current dataset is sufficient for the application.

**Additional Feedback:**

Can you comment about the points discussed in the weakness section?   The main concern I have is that the dataset may be more of academic interest given that real-world machine vision systems will employ alternative strategies for classification of material types.  Moreover, can you comment about how this dataset helps the robotics researchers?

**Clarity:**

The paper is well written although the paper is redundant (there is significant overlap between the main body text and the answers for the checklist).

**Correctness:**

See comments above on weakness.
The evaluation methods and experiment design is at a baseline level preliminary and could be strengthened.  Its hard to judge from the figures in the supplemental (e.g. figures 9, 12) what really the current baselines are successful at. There is no discussion of the specifics of failures.  I would recommend iterating and improving the baseline to provide adequate insights into the achievable performance using state of the art techniques.

**Documentation:**

There is sufficient detail on the data collection, organization, availability and maintenance.  The licensing issue as well as hosting and maintenance is addressed.  Code for the experiments is provided along with the datasets to support reproduction of results.

**Ethics:**

None.

**Relation To Prior Work:**

Adequate discussion of past datasets relevant to this applicaton is presented and there are statements on how their dataset differs from the previous datasets.

**Summary And Contributions:**

This paper describes a dataset acquisition and annotation protocol for a waste recycling task.  Images are obtained from a real-world industrial waste setup wherein waste material (including paper, brown paper, cardboard, soft / rigid plastic ) are imaged as they arrive in a conveyer belt using a camera setup.  Several datasets are provided to address fully supervised segmentation, semi-supervised segmentation, weakly supervised segmentation of object classes and binary classification of frames before and after material sorting tasks.   The contribution of the dataset is that it fills the void in realistic datasets for waste sorting task.  In addition, well known algorithms are used to demonstrate baseline performance and to illustrate that there is a significant need for performance improvement.

---

> ### Author Response · Authors · 2021-07-09
> **Reply to Reviewer Criz**
>
> We thank the reviewer for a thorough and thoughtful review. Below, we address the main concerns highlighted in the review.
> 1. *No NIR data is used for material prediction*. While it is true that NIR, spectral sensors and other specific sensors have been used to separate specific material types, such as types of plastic [https://www.sciencedirect.com/science/article/pii/S2542504818300113] or metal [https://www.mdpi.com/1424-8220/19/2/247], we stress that: a) our task is slightly different, as we do not need to separate HDPA from other types of plastic, or aluminum from copper. Our goal is to separate cardboard, metal and plastic from paper. For this specific task, the RGB signal suffices, and actually provides more information. Humans can easily solve this task using eye perception, and we are providing a benchmark that would catalyze computer vision systems achieving this goal;  b) most industrial companies that develop waste sorting robots, such as AMP Robotics,  Waste Robotics, Zen Robotics etc., only use the RGB signal which suggests that it is sufficient for waste sorting; c) We understand that while RGB signals are sufficient to solve the given task, NIR footage modality can provide some additional information that may increase the prediction accuracy. For the community interested in multi-modal research, we plan to provide the NIR data along with the main dataset as well.
> 2. *Limited usefulness to robotics practitioners*. Our main goal is to develop an in-the-wild robotic grasping system that automatically detects and grasps waste of specific types from the conveyor belt. Grasping potentially deformable objects in cluttered environments is a challenging and important problem in robotics research. In order to learn how to grasp objects efficiently in an **unrestricted environment**, a robotic system must be able to detect the location, shape and rigidity of the waste object on the fly during training. For that, a segmentation method is needed,  which requires some annotated data for training. Therefore, our dataset is an essential component in the development of the robotic waste sorting system. Our dataset is unique and valuable to the robotics research community because it is the first in-the-wild waste dataset with cluttered scenes and separation of soft and rigid object types. Our dataset can be used for training and evaluation of the computer vision methods, which are a prerequisite for training a robotics system. We would like to add that other RGB datasets with segmentation, such as Cornell Grasping Dataset, are widely used in the robotics research community in a significantly more controlled environment.
> 3. *Sampling strategy not clear*. The belt speed is changed manually based on how much material is on the belt to allow workers to manually pick up materials, therefore it is impossible to exactly estimate the redundancy. It is similarly hard to estimate the redundancy in other  in-the-wild video segmentation datasets, such as Cityscapes or DAVIS.
> 4. *Dataset is too small*. While, indeed, most popular vision datasets count in millions these days, full annotation of our data is very expensive due to the challenging nature of the task. We stress that we are proposing *the largest* public waste sorting dataset with densely annotated objects. Since in our setup, as in most real applications, full annotation is extremely difficult and expensive and therefore the fully-supervised methods are not a viable option, our aim is to challenge the computer vision community to develop more data-efficient algorithms that utilize unlabeled and weakly-labeled data. In this paper, we demonstrated that none of the existing state-of-the-art methods, including fully-, semi- and weakly-supervised methods, can solve this real-world problem.
> 5. *Baselines too weak & no discussion*. We chose current state-of-the-art methods for each specific task (detection, semantic segmentation), and we believe this is a reasonable choice for a novel dataset. For each task, we conducted a set of more than 20 experiments with different hyperparameters and reported the best results and optimal hyperparameters in the paper. The fact that these methods perform poorly on the proposed dataset indicates their limitations  discussed in the Results (L 221-230 for detection, L 267-279 for segmentation) paragraph for each experiment, that we will quickly reiterate here: a) fully-supervised methods, such as MaskRCNN and DeepLabv3+, overfit to the relatively small training set and perform poorly on the least frequent classes (see class-wise results and confusion matrices  in Section B.1 of Appendix); 2) introduction of unlabeled examples in the semi-supervised CCT method improves the segmentation accuracy of the most frequent classes, but suppresses the prediction least frequent classes (see Section B2 of the Appendix for class-wise results, and Figure 12).  We will move the most important results from the Appendix to the main paper.

---

> > ### Comment · Reviewer_Criz · 2021-07-13
> > **Questions posed were answered, but still unconvinced**
> >
> > Thank you very much for the detailed comments.  I understand that you believe that this dataset is valuable dataset in the specific application domain and it provides a challenging benchmark for computer vision and robotics research.  In my humble opinion, in its present form, the dataset is not useful for robotics research.  Moreover, the awareness that this domain is complex for computer vision baselines is also not necessarily novel.  I do agree that annotation is costly and therefore the dataset size is perhaps from an academician's standpoint reasonable (moreover your point that this promotes development of algorithms that work with small sample sizes is indeed potentially true).   I feel that your work can be strengthened by adding very specific visual + IR + 3D data that can spur interesting robotic vision research for waste management.  I stick with my original rating as I am not convinced that the paper is ready for publication.

---

### Decision · Program_Chairs · 2021-07-26

**Decision:**

Reject

**Comment:**

Reviewers agree on the importances of this dataset. However, a few more iterations may be needed for this to be ready for publications - we recommend authors consider submitting the second round with these revisions. To mention a few, 1) clarifying how “other ways” to do the same task (e.g., NIR data) is relevant, pros and cons 2) clarify how the proposed applications are relevant e.g., robotic grasping 2) [important] discuss failure modes to provide insights 4) other computational aspects: iterate and improve baselines, clarify sampling strategy etc.